# WAVELET POOLING FOR CONVOLUTIONAL NEURAL NETWORKS

**Travis Williams**
Department of Electrical Engineering
North Carolina A&T State University
Greensboro, NC 27410, USA
tlwilli3@aggies.ncat.edu

**Robert Li**
Department of Electrical Engineering
North Carolina A&T State University
Greensboro, NC 27410, USA
eeli@ncat.edu

## ABSTRACT

Convolutional Neural Networks continuously advance the progress of 2D and 3D image and object classification. The steadfast usage of this algorithm requires constant evaluation and upgrading of foundational concepts to maintain progress. Network regularization techniques typically focus on convolutional layer operations, while leaving pooling layer operations without suitable options. We introduce Wavelet Pooling as another alternative to traditional neighborhood pooling. This method decomposes features into a second level decomposition, and discards the first-level subbands to reduce feature dimensions. This method addresses the overfitting problem encountered by max pooling, while reducing features in a more structurally compact manner than pooling via neighborhood regions. Experimental results on four benchmark classification datasets demonstrate our proposed method outperforms or performs comparatively with methods like max, mean, mixed, and stochastic pooling.

## 1 INTRODUCTION

Convolutional Neural Networks (CNNs) have become the standard-bearer in image and object classification (Nielsen, 2015). Due to the layer structures conforming to the shape of the inputs, CNNs consistently classify images, objects, videos, etc. at a higher accuracy rate than vector-based deep learning techniques (Nielsen, 2015). The strength of this algorithm motivates researchers to constantly evaluate and upgrade foundational concepts to continue growth and progress. The key components of CNN, the convolutional layer and pooling layer, consistently undergo modifications and innovations to elevate accuracy and efficiency of CNNs beyond previous benchmarks.

Pooling has roots in predecessors to CNN such as Neocognitron, which manual subsampling by the user occurs (Fukushima, 1979), and Cresceptron, which introduces the first max pooling operation in deep learning (Weng et al., 1992). Pooling subsamples the results of the convolutional layers, gradually reducing spatial dimensions of the data throughout the network. The benefits of this operation are to reduce parameters, increase computational efficiency, and regulate overfitting (Boureau et al., 2010).

Methods of pooling vary, with the most popular form being max pooling, and secondarily, average pooling (Nielsen, 2015; Lee et al., 2016). These forms of pooling are deterministic, efficient, and simple, but have weaknesses hindering the potential for optimal network learning (Lee et al., 2016; Yu et al., 2014). Other pooling operations, notably mixed pooling and stochastic pooling, use probabilistic approaches to correct some of the issues of the prior methods (Yu et al., 2014; Zeiler & Fergus, 2013).

However, one commonality all these pooling operations employ a neighborhood approach to subsampling, reminiscent of nearest neighbor interpolation in image processing. Neighborhood interpolation techniques perform fast, with simplicity and efficiency, but introduce artifacts such as edge halos, blurring, and aliasing (Parker et al., 1983). Minimizing discontinuities in the data are critical to aiding in network regularization, and increasing classification accuracy.

We propose a wavelet pooling algorithm that uses a second-level wavelet decomposition to subsample features. Our approach forgoes the nearest neighbor interpolation method in favor of an organic, subband method that more accurately represents the feature contents with less artifacts. We compare our proposed pooling method to max, mean, mixed, and stochastic pooling to verify its validity, and ability to produce near equal or superior results. We test these methods on benchmark image classification datasets such as Mixed National Institute of Standards and Technology (MNIST) (Lecun et al., 1998), Canadian Institute for Advanced Research (CIFAR-10) (Krizhevsky, 2009), Street House View Numbers (SHVN) (Netzer et al., 2011), and Karolinska Directed Emotional Faces (KDEF) (Lundqvist et al., 1998). We perform all simulations in MATLAB R2016b.

The rest of this paper organizes as follows: Section 2 gives the background, Section 3 describes the proposed methods, Section 4 discusses the experimental results, and Section 5 gives the summary and conclusion.

## 2 BACKGROUND

Pooling is another term for subsampling. In this layer, the dimensions of the output of the convolutional layer are condensed. The dimensionality reduction happens by summarizing a region into one neuron value, and this occurs until all neurons have been affected. The two most popular forms of pooling are max pooling and average pooling (Nielsen, 2015; Lee et al., 2016). Max pooling involves taking the maximum value of a region $R_{ij}$ and selecting it for the condensed feature map. Average pooling involves calculating the average value of a region and selecting it for the condensed feature map. The max pooling function is expressed as:

$$a_{kij} = \max_{(p,q) \in R_{ij}} (a_{kpq})$$
(1)

While average pooling is shown by the following equation:

$$a_{kij} = \frac{1}{|R_{ij}|} \sum_{(p,q) \in R_{ij}} a_{kpq}$$
(2)

Where $a_{kij}$ is the output activation of the $k^{th}$ feature map at *(i,j)*, $a_{kpq}$ is the input activation at *(p,q)* within $R_{ij}$, and $|R_{ij}|$ is the size of the pooling region. An illustration of both of these pooling methods is expressed in Figure 1 (Williams & Li, 2016):

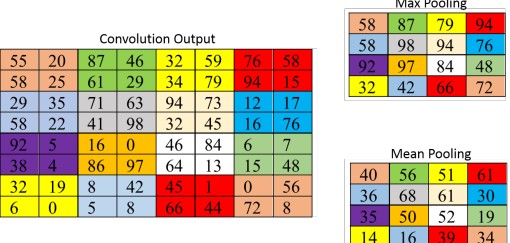

Figure 1: Example of Max & Average Pooling with Stride of 2

While max and average pooling both are effective, simple methods, they also have shortcomings. Max pooling, depending on the data, can erase details from an image (Yu et al., 2014; Zeiler & Fergus, 2013). This happens if the main details have less intensity than the insignificant details. In addition, max pooling commonly overfits training data (Yu et al., 2014; Zeiler & Fergus, 2013). Average pooling, depending on the data, can dilute pertinent details from an image. The averaging of data with values much lower than significant details causes this action (Yu et al., 2014; Zeiler & Fergus, 2013). Figure 2 illustrates these shortcomings using the toy image example:

To combat these issues, researchers have created probabilistic pooling methods. Mixed pooling combines max and average pooling by randomly selecting one method over the other during training

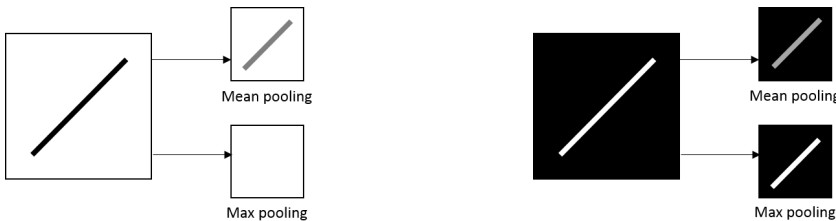

Figure 2: Shortcomings of Max & Average Pooling using Toy Image

(Yu et al., 2014). There is no set way to perform mixed pooling. This method is applied arbitrarily in three different ways (1) for all features within a layer, (2) mixed between features within a layer, or (3) mixed between regions for different features within a layer (Lee et al., 2016; Yu et al., 2014). Mixed pooling is shown in the following equation:

$$a_{kij} = \lambda \cdot \max_{(p,q) \in R_{ij}} (a_{kpq}) + (1 - \lambda) \cdot \frac{1}{|R_{ij}|} \sum_{(p,q) \in R_{ij}} a_{kpq} \tag{3}$$

where $\lambda$ is a random value 0 or 1, indicating max or average pooling for a particular region/feature/layer.

Another probabilistic pooling method, called stochastic pooling, improves upon max pooling by randomly sampling from neighborhood regions based on the probability values of each activation (Zeiler & Fergus, 2013). These probabilities $p$ for each region are calculated by normalizing the activations within the region:

$$p_{pq} = \frac{a_{pq}}{\sum_{(p,q) \in R_{ij}} a_{pq}} \tag{4}$$

The pooled activation is sampled from a multinomial distribution based on $p$ to pick a location $l$ within the region (Zeiler & Fergus, 2013). The process is captured in the following equation (Zeiler & Fergus, 2013):

$$a_{kij} = a_l \quad where \quad l \sim P(p_1, ..., p_{|R_{ij}|}) \tag{5}$$

Figure 3 displays a visual example of stochastic pooling on a 3x3 region:

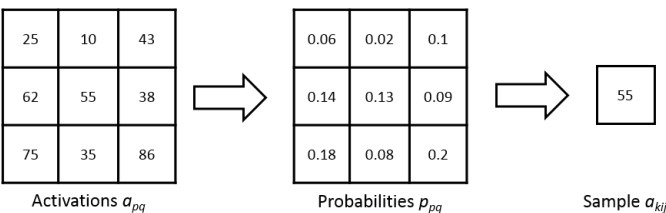

Figure 3: Stochastic Pooling Example

In Figure 3, a region of activations are shown, and in the adjacent region, their corresponding probabilities based on Equation 4. In any given region, the activations with the highest probabilities have the higher chance of selection. However, any activation can be chosen. In this example, the stochastic pooling method selects the midrange activation with a probability of 13%. By being based off probability, and not deterministic, stochastic pooling avoids the shortcomings of max and average pooling, while enjoying some of the advantages of max pooling (Zeiler & Fergus, 2013).

## 3 Proposed Method

The previously highlighted pooling methods use neighborhoods to subsample, almost identical to nearest neighbor interpolation. Previous studies explore the possibilities of wavelets in image interpolation versus traditional methods (Dumic et al., 2007). Our proposed pooling method uses wavelets to reduce the dimensions of the feature maps. We propose using the wavelet transform to minimize artifacts resulting from neighborhood reduction (Parker et al., 1983). We postulate that our approach, which discards the first-order subbands, more organically captures the data compression. This organic reduction therefore lessens the creation of jagged edges and other artifacts that may impede correct image classification.

### 3.1 Forward Propagation

The proposed wavelet pooling scheme pools features by performing a 2nd order decomposition in the wavelet domain according to the fast wavelet transform (FWT) (Mallat, 1989; Nason & Silverman, 1995; Strang & Nguyen, 1996; Burrus et al., 1998), which is a more efficient implementation of the two-dimensional discrete wavelet transform (DWT) as follows (Chui, 1992; Strang & Strela, 1995; Rieder et al., 1994):

$$W_\varphi[j+1,k] = h_\varphi[-n] * W_\varphi[j,n]|_{n=2k,k\leq 0} \tag{6}$$

$$W_\psi[j+1,k] = h_\psi[-n] * W_\psi[j,n]|_{n=2k,k\leq 0} \tag{7}$$

where $\varphi$ is the approximation function, and $\psi$ is the detail function, $W_\varphi$, $W_\psi$ are called approximation and detail coefficients. $h_\varphi[-n]$ and $h_\psi[-n]$ are the time reversed scaling and wavelet vectors, *(n)* represents the sample in the vector, while *(j)* denotes the resolution level. When using the FWT on images, we apply it twice (once on the rows, then again on the columns). By doing this in combination, we obtain our detail subbands (LH, HL, HH) at each decomposition level, and our approximation subband (LL) for the highest decomposition level.

After performing the 2nd order decomposition, we reconstruct the image features, but only using the 2nd order wavelet subbands. This method pools the image features by a factor of 2 using the inverse FWT (IFWT) (Mallat, 1989; Nason & Silverman, 1995; Strang & Nguyen, 1996; Burrus et al., 1998), which is based off of the inverse DWT (IDWT) (Chui, 1992; Strang & Strela, 1995; Rieder et al., 1994):

$$W_\varphi[j,k] = h_\varphi[-n] * W_\varphi[j+1,n] + h_\psi[-n] * W_\psi[j+1,n]|_{n=\frac{k}{2},k\leq 0} \tag{8}$$

Figure 4 gives an illustration of the algorithm for the forward propagation of wavelet pooling:

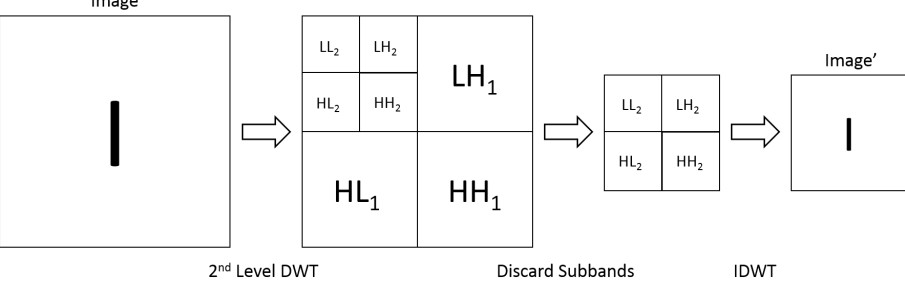

Figure 4: Wavelet Pooling Forward Propagation Algorithm

### 3.2 BACKPROPAGATION

The proposed wavelet pooling algorithm performs backpropagation by reversing the process of its forward propagation. First, the image feature being back propagated undergoes $1^{st}$ order wavelet decomposition. After decomposition, the detail coefficient subbands upsample by a factor of 2 to create a new $1^{st}$ level decomposition. The initial decomposition then becomes the $2^{nd}$ level decomposition. Finally, this new $2^{nd}$ order wavelet decomposition reconstructs the image feature for further backpropagation using the IDWT. Figure 5 details the backpropagation algorithm of wavelet pooling:

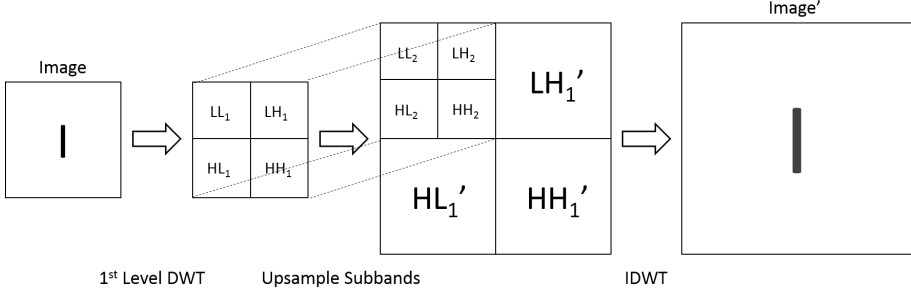

Figure 5: Wavelet Pooling Backpropagation Algorithm

## 4 RESULTS AND DISCUSSION

All CNN experiments use MatConvNet (Vedaldi & Lenc, 2015). All training uses stochastic gradient descent (Bottou, 2010). For our proposed method, the wavelet basis is the Haar wavelet, mainly for its even, square subbands. All experiments are run on a 64-bit operating system, with an Intel Core i7-6800k CPU @ 3.40 GHz processor, with 64.0 GB of RAM. We utilize two GeForce Titan X Pascal GPUs with 12 GB of video memory for all training. All CNN structures except for MNIST use a network loosely based on Zeilers network (Zeiler & Fergus, 2013). We repeat the experiments with Dropout (Srivastava, 2013) and replace Local Response Normalization (Krizhevsky, 2009) with Batch Normalization (Ioffe & Szegedy, 2015) for CIFAR-10 and SHVN (Dropout only) to examine how these regularization techniques change the pooling results. To test the effectiveness of each pooling method on each dataset, we solely pool with that method for all pooling layers in that network. All pooling methods use a 2x2 window for an even comparison to the proposed method. Figure 6 gives a selection of each of the datasets.

### 4.1 MNIST

The network architecture is based on the example MNIST structure from MatConvNet, with batch normalization inserted. All other parameters are the same. Figure 7 shows our network structure for the MNIST experiments:

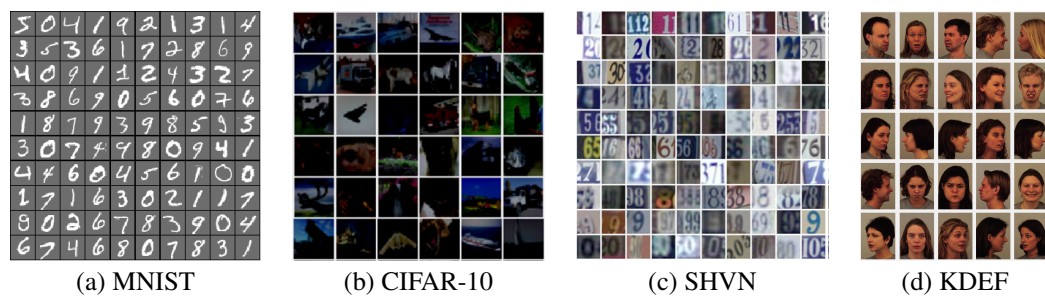

| (a) MNIST | (b) CIFAR-10 | (c) SHVN | (d) KDEF |

Figure 6: Selection of Image Datasets

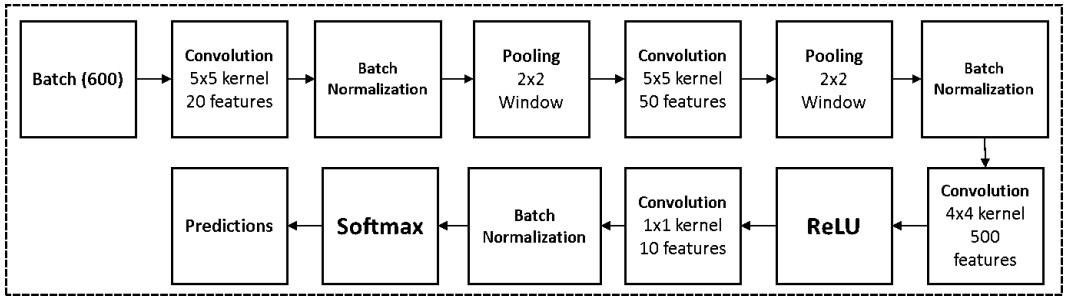

Figure 7: CNN MNIST Structure Block Diagram

The input training data and test data come from the MNIST database of handwritten digits. The full training set of 60,000 images is used, as well as the full testing set of 10,000 images. Table 1 shows our proposed method outperforms all methods. Given the small number of epochs, max pooling is the only method to start to overfit the data during training. Mixed and stochastic pooling show a rocky trajectory, but do not overfit. Average and wavelet pooling show a smoother descent in learning and error reduction. Figure 8 shows the energy of each method per epoch.

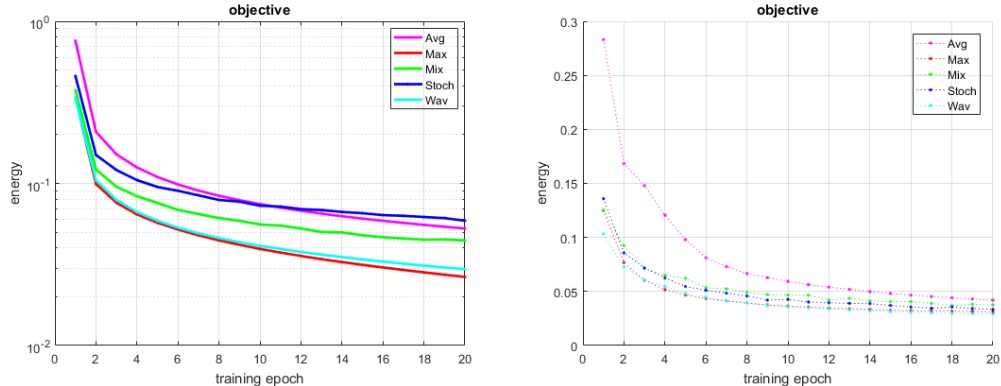

Figure 8: MNIST Pooling Method Energy Performance of Training & Validation Sets

Table 1 shows the accuracy of each method:

|  | Average | Max | Mixed | Stochastic | Wavelet |
|---|---|---|---|---|---|
| Accuracy (%) | 98.72 | 98.80 | 98.86 | 98.90 | **99.01** |

Table 1: MNIST Performance of Pooling Methods

## 4.2 CIFAR-10

We run two sets of experiments with the pooling methods. The first is a regular network structure with no dropout layers. We use this network to observe each pooling method without extra regularization. The second uses dropout and batch normalization, and performs over 30 more epochs to observe the effects of these changes. Figure 9 shows our network structure for the CIFAR-10 experiments:

The input training and test data come from the CIFAR-10 dataset. The full training set of 50,000 images is used, as well as the full testing set of 10,000 images. For both cases, with no dropout, and with dropout, Table 2 and Table 3 show our proposed method has the second highest accuracy. Max pooling overfits fairly quickly, while wavelet pooling resists overfitting. The change in learning rate prevents our method from overfitting, and it continues to show a slower propensity for learning. Mixed and stochastic pooling maintain a consistent progression of learning, and their validation sets

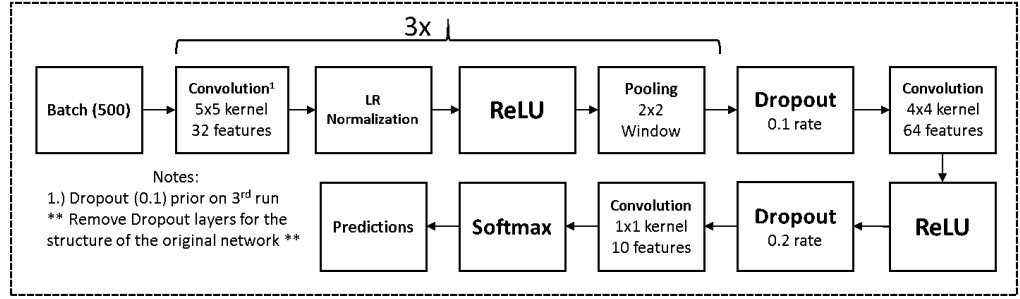

Figure 9: CNN CIFAR-10 Structure Block Diagram

trend at a similar, but better rate than our proposed method. Average pooling shows the smoothest descent in learning and error reduction, especially in the validation set. Figure 10 shows the energy of each method per epoch.

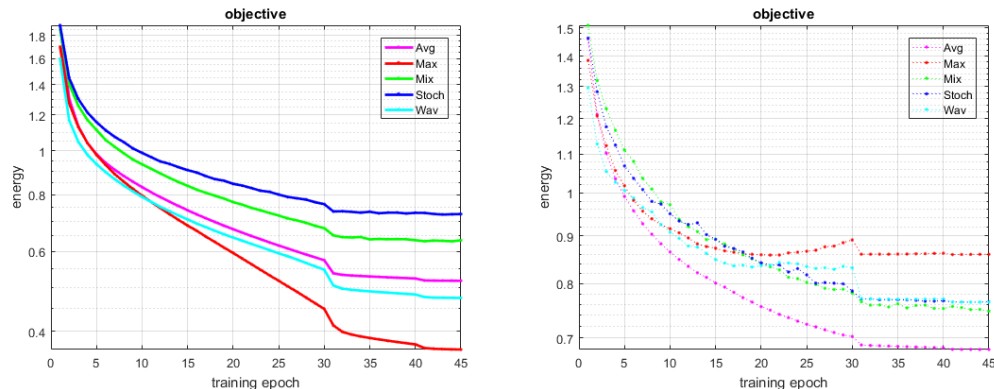

Figure 10: CIFAR-10 Pooling Method Energy Performance of Training & Validation Sets

Tables 2 and 3 show the accuracy of each method:

|  | Average | Max | Mixed | Stochastic | Wavelet |
|---|---|---|---|---|---|
| Accuracy (%) | **76.51** | 71.42 | 73.77 | 73.03 | 74.42 |

Table 2: CIFAR-10 Performance of Pooling Methods

|  | Average | Max | Mixed | Stochastic | Wavelet |
|---|---|---|---|---|---|
| Accuracy (%) | **81.15** | 80.30 | 79.21 | 80.09 | 80.28 |

Table 3: CIFAR-10 Performance of Pooling Methods + Dropout

## 4.3 SHVN

We run two sets of experiments with the pooling methods. The first is a regular network structure with no dropout layers. We use this network to observe each pooling method without extra regularization. The second uses dropout to observe the effects of this change. Figure 11 shows our network structure for the SHVN experiments:

The input training and test data come from the SHVN dataset. For the case with no dropout, we use 55,000 images from the training set. For the case with dropout, we use the full training set of 73,257 images, a validation set of 30,000 images we extract from the extra training set of 531,131 images, as well as the full testing set of 26,032 images. For both cases, with no dropout, and with dropout,

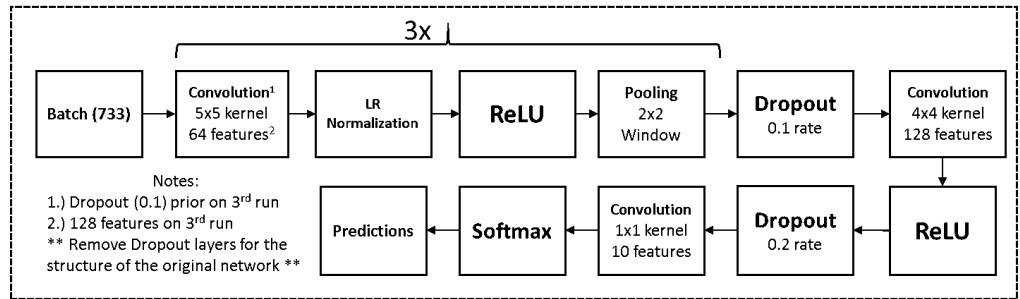

Figure 11: CNN SHVN Structure Block Diagram

Table 4 and Table 5 show our proposed method has the second lowest accuracy. Max and wavelet pooling both slightly overfit the data. Our method follows the path of max pooling, but performs slightly better in maintaining some stability. Mixed, stochastic, and average pooling maintain a slow progression of learning, and their validation sets trend at near identical rates. Figure 12 shows the energy of each method per epoch.

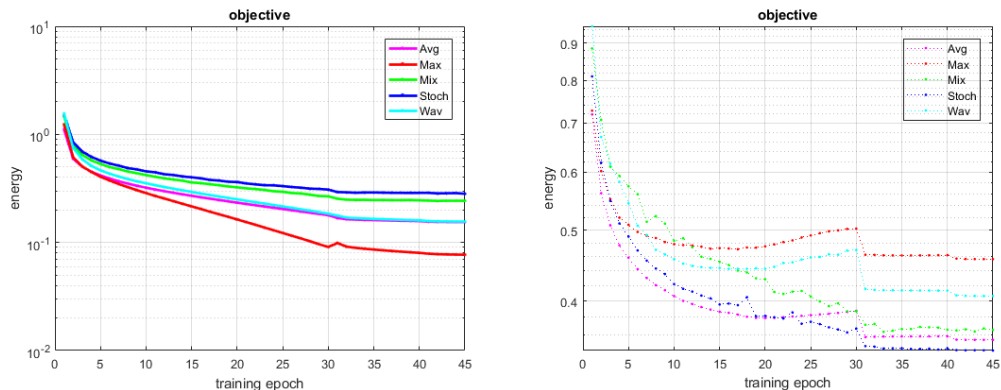

Figure 12: SHVN Pooling Method Energy Performance of Training & Validation Sets

Tables 4 and 5 shows the accuracy of each method:

|  | Average | Max | Mixed | Stochastic | Wavelet |
|---|---|---|---|---|---|
| Accuracy (%) | 89.83 | 88.09 | 89.25 | **89.97** | 88.51 |

Table 4: SHVN Performance of Pooling Methods

|  | Average | Max | Mixed | Stochastic | Wavelet |
|---|---|---|---|---|---|
| Accuracy (%) | **92.80** | 92.18 | 92.13 | 91.04 | 91.10 |

Table 5: SHVN Performance of Pooling Methods + Dropout

## 4.4 KDEF

We run one set of experiments with the pooling methods that includes dropout. Figure 13 shows our network structure for the KDEF experiments:

The input training and test data come from the KDEF dataset. This dataset contains 4,900 images of 35 people displaying seven basic emotions (afraid, angry, disgusted, happy, neutral, sad, and surprised) using facial expressions. They display emotions at five poses (full left and right profiles, half left and right profiles, and straight).

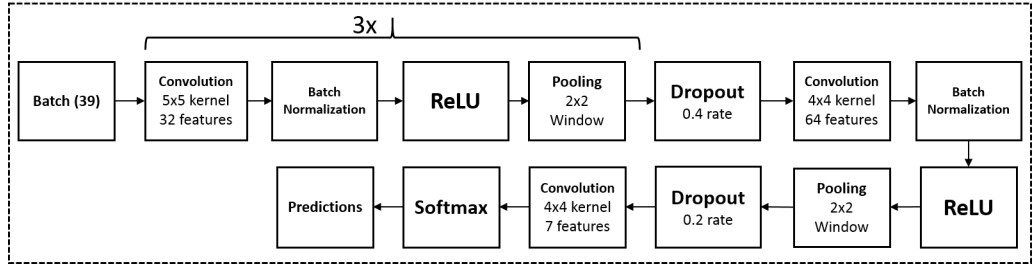

Figure 13: CNN KDEF Structure Block Diagram

This dataset contains a few errors that we fix (missing or corrupted images, uncropped images, etc.). All of the missing images are at angles of -90, -45, 45, or 90 degrees. We fix the missing and corrupt images by mirroring their counterparts in MATLAB and adding them back to the dataset. We manually crop the images that need to match the dimensions set by the creators (762 x 562). KDEF does not designate a training or test data set. We shuffle the data and separate 3,900 images as training data, and 1,000 images as test data. We resize the images to 128x128 because of memory and time constraints.

The dropout layers regulate the network and maintain stability in spite of some pooling methods known to overfit. Table 6 shows our proposed method has the second highest accuracy. Max pooling eventually overfits, while wavelet pooling resists overfitting. Average and mixed pooling resist overfitting, but are unstable for most of the learning. Stochastic pooling maintains a consistent progression of learning. Wavelet pooling also follows a smoother, consistent progression of learning. Figure 14 shows the energy of each method per epoch.

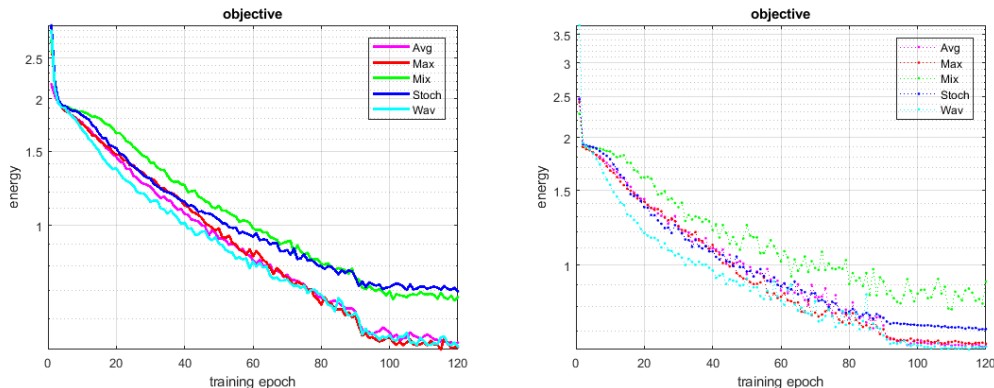

Figure 14: KDEF Pooling Method Energy Performance of Training & Validation Sets

Table 6 shows the accuracy of each method:

|  | Average | Max | Mixed | Stochastic | Wavelet |
|---|---|---|---|---|---|
| Accuracy (%) | **76.5** | 75.6 | 72.6 | 72.7 | 75.9 |

Table 6: KDEF Performance of Pooling Methods + Dropout

## 4.5 COMPUTATIONAL COMPLEXITY

Our construction and implementation of wavelet pooling is not efficient. We present this proposed methods as a proof-of-concept, to show its potential and validity, and also to be open to massive improvements. The main area of improvement is computational efficiency. As a proof-of-concept, the code written to implement this method is not at its peak form. Additionally, we did not have

the time, space, or resources to optimize the code. We view the accuracy results and novelty as a starting point to spawn improvements, both from our own research as well as other researchers.

We calculate efficiency in terms of mathematical operations (multiplications, additions, logical, etc.) that each method utilizes to complete its algorithm. For max pooling, we calculate operations based on the worst-case scenarios for each neighborhood in finding the maximum value. For average pooling, we calculate the number of additions and division for each neighborhood. Mixed pooling is the mean value of both average and max pooling. We calculate operations for stochastic pooling by counting the number of mathematical operations as well as the random selection of the values based on probability (Roulette Wheel Selection). For wavelet pooling, we calculate the number of operations for each subband at each level, in both decomposition and reconstruction.

Table 7 shows the number of mathematical operations for one image in forward propagation. This table shows that for all methods, average pooling has the least number of computations, followed by mixed pooling, with max pooling not far behind. Stochastic pooling is the least computationally efficient pooling method out of the neighborhood-based methods. It uses about 3x more mathematical operations than average pooling, the most computationally efficient.

However, wavelet pooling by far is the least computationally efficient method, using 54 to 213x more mathematical operations than average pooling. This is partially due to the implementation of the subband coding, which did not implement multidimensional decomposition and reconstruction.

|       | MNIST | CIFAR-10 | SHVN | KDEF |
|-------|-------|----------|------|------|
| Max   | 6.2K  | 13K      | 26K  | 50K  |
| Avg   | **3.5K** | **7.4K** | **15K** | **29K** |
| Mix   | 4.8K  | 10K      | 20K  | 40K  |
| Stoch | 10.6K | 22K      | 45K  | 86K  |
| Wav   | 110K  | 405K     | 810K | 6.2M |

Table 7: Number of Mathematical Operations for Each Method According to Dataset

Nonetheless, by implementing our method through good coding practices (vectorization, architecture, etc.), GPUs, and an improved FTW algorithm, this method can prove to be a viable option. There exists a few improvements to the FTW algorithm that utilize multidimensional wavelets (Karlsson & Vetterli, 1988; Weeks & Bayoumi, 1998), lifting (Valens, 1999), parallelization Holmström (1995), as well as other methods that boast of improving the efficiency in speed and memory (Oliver & Malumbres, 2008; Khoromskij & Miao, 2014; Kopenkov, 2008)

## 5   CONCLUSION

We prove wavelet pooling has potential to equal or eclipse some of the traditional methods currently utilized in CNNs. Our proposed method outperforms all others in the MNIST dataset, outperforms all but one in the CIFAR-10 and KDEF datasets, and performs within respectable ranges of the pooling methods that outdo it in the SHVN dataset. The addition of dropout and batch normalization show our proposed methods response to network regularization. Like the non-dropout cases, it outperforms all but one in both the CIFAR-10 & KDEF datasets, and performs within respectable ranges of the pooling methods that outdo it in the SHVN dataset. Our results confirm previous studies proving that no one pooling method is superior, but some perform better than others depending on the dataset and network structure Boureau et al. (2010); Lee et al. (2016). Furthermore, many networks alternate between different pooling methods to maximize the effectiveness of each method.

Future work and improvements in this area could be to vary the wavelet basis to explore which basis performs best for the pooling. Altering the upsampling and downsampling factors in the decomposition and reconstruction can lead to better image feature reductions outside of the 2x2 scale. Retention of the subbands we discard for the backpropagation could lead to higher accuracies and fewer errors. Improving the method of FTW we use could greatly increase computational efficiency. Finally, analyzing the structural similarity (SSIM) of wavelet pooling versus other methods could further prove the vitality of using our approach.

ACKNOWLEDGMENTS

This research is supported by the Title III HBGI PhD Fellowship grant from the U.S. Department of Education.

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
