# OpenReview forum: "Wavelet Pooling for Convolutional Neural Networks"
_ICLR.cc/2018/Conference — Accept (Poster)_

### Official Review · AnonReviewer2 · 2017-11-23
**Well written/motivated paper, some extra polishing would make it solid.**

**Rating:** 7
**Confidence:** 4

**Review:**

The paper proposes "wavelet pooling" as an alternative for traditional subsampling methods, e.g. max/average/global pooling, etc., within convolutional neural networks.
Experiments on the MNIST, CIFAR-10, SHVN and KDEF datasets, shows the proposed wavelet-based method has
competitive performance with existing methods while still being able to address the overfitting behavior of max pooling.

Strong points
- The method is sound and well motivated.
- The proposes method achieves competitive performance.

Weak points
- No information about added computational costs is given.
- Experiments are conducted in relatively low-scale datasets.


Overall the method is well presented and properly motivated. The paper as a good flow and is easy to follow. The authors effectively demonstrate with few toy examples the weaknesses of traditional methods, i.e max pooling and average pooling. Moreover, their extended evaluation on several datasets show the performance of the proposed method in different scenarios.

My main concerns with the manuscript are the following.

Compared to traditional methods, the proposed methods seems to require higher computation costs. In a deep neural network setting where operations are conducted a large number of times, this is a of importance. However, no indication is given on what are the added computation costs of the proposed method and how that compares to existing methods. A comparison on that regard would strengthen the paper.

In many of the experiments, the manuscript stresses the overfitting behavior of max pooling. This makes me wonder whether this is caused by the fact that experiments are conducted or relatively smaller datasets. While the currently tested datasets are a good indication of the performance of the proposed method, an evaluation on a large scale scenario, e.g. ILSVRC'12, could solidify the message sent by this manuscript. Moreover, it would increase the relevance of this work in the computer vision community.

Finally, related to the presentation, I would recommend presenting the plots, i.e. Fig. 8,10,12,14, for the training and validation image subsets in two separate plots. Currently, results for training and validation sets are mixed in the same plot, and due to the clutter it is not possible to see the trends clearly.
Similarly, I would recommend referring to the Tables added in the paper when discussing the performance of the proposed method w.r.t. traditional alternatives.

I encourage the authors to address my concerns in their rebuttal

---

> ### Author Response · Authors · 2017-12-17
> **Re: Well written/motivated paper, some extra polishing would make it solid.**
>
> ** Update ** I have not been able to run experiments or add computational costs, etc due to varying life factors (i.e. graduation & packing/moving for my postdoc in such a short window of time).
>
> Thank you for your review. I will address your main concerns below with our paper.
>
> Computation costs:
> - You are correct, this method does require higher computational costs
> - In our initial implementation of this method, we didn't employ advanced programming methods and pre-compiling that would speed up the computations and reduce the number of operations. In the future these will be integrated to ensure usability.
> - Our method uses FWT (Fast Wavelet transform) which is much faster than DWT. O(N) versus O(k*2^n)
> - I will add a comparison based on mathematical operations and an explanation on how to lessen these costs
>
> Max pooling overfitting:
> - It is a strong possibility that the nature of the datasets contributes to max pooling overfitting faster.
> - However, in various papers we surveyed, this conclusion was reached because of the nature of the algorithm.
> - I do suspect therefore that in a larger dataset this trend would still be true, but perhaps not as fast.
>
> Larger Datasets:
> - I agree that testing on a larger dataset would remove all doubt.
> - I will apply this method to a larger dataset, but I am not sure it will make it into this paper, or be reserved for another manuscript.
>
> Presentation:
> - I will redo the plots to fit the manner you described
> - I will reference the tables when discussing the performance of the proposed wrt traditional alternatives

---

> ### Author Response · Authors · 2018-02-19
> **Re: Well written/motivated paper, some extra polishing would make it solid.**
>
> We were actually able to add the computational complexity component as a subsection within the results and discussion. We also modified the graphs and referenced the tables as you requested.

---

### Official Review · AnonReviewer1 · 2017-11-25
**interesting idea for "pooling"**

**Rating:** 9
**Confidence:** 3

**Review:**

I think this paper presents an interesting take on feature pooling. In particular, the idea is to look at pooling as some form of a lossy process, and try to find such a process such that it discards less information given some decimation criterion. Once formulating the problem like this, it becomes obvious that wavelets are a very good candidate.

Pros:
- The nice thing about this method is that average pooling is in some sense a special case of this method, so we can see a clear connection.
- Lots of experiments, and results, which show the method both performing the best in some cases, and not the best in others. I applaud the authors for publishing all the experiments they ran because some may have been tempted to "forget" about the experiments in which the proposed method did not perform the best.

Cons:
- No comparison to non-wavelet methods. For example, one obvious comparison would have been to look at using a DCT or FFT transform where the output would discard high frequency components (this can get very close to the wavelet idea!).
- This method has the potential to show its potential on larger pooling windows than 2x2. I would have loved to see some experiments that prove/disprove this.

Other comments:
- Given that this method's flexibility, I could imagine this generate a new class of pooling methods based on lossy transforms. For example, given a MxNxK input, the wavelet idea can be made to output (M/D)x(N/D)x(K/D) (where D is decimation factor). Of interest is the fact that channels can be treated just like any other dimension, since information will be preserved!

Final comments:
- I like the idea and it seems novel it may lead to some promising research directions related to lossy pooling methods/channel aggregation. As such, I think it will be a nice addition to ICLR, especially if the authors decide to run some of the experiments I was suggesting, namely: show what happens when larger pooling windows are used (say 4x4 instead of 2x2), and compare to other lossy techniques (such as Fourier or cosine-transforms).

---

> ### Author Response · Authors · 2017-12-17
> **Re: interesting idea for "pooling"**
>
> ** Update ** I have not been able to run experiments or add computational costs, etc due to varying life factors (i.e. graduation & packing/moving for my postdoc in such a short window of time).
>
> Thank you for your review. I agree with everything you mentioned in your pros, cons, and other commentary. I tried to show the whole spectrum of our results so that we could show integrity, and a lane for improvement into this initial idea. I will briefly address some of the points you made as well.
>
> Non-wavelet methods:
> - I wholeheartedly agree with comparing the DWT method to DCT, FFT, etc.
> - I didn't implement these for this paper because I wanted to focus on the effect of the DWT
> - I can see a pathway for a journal or another paper comparing DWT, DCT, FFT, etc style methods to traditional methods
>
> Larger pooling windows:
> - I agree, the experiments should have included another window size
> - Initially I wanted to see the performance on the 1st level (2x2) before reevaluating.
> - I will run a comparison on 4x4!
>
> I also can see a lane for further research into lossy pooling/channel aggregation, and I hope to be a contributor. I will try to compare to the other lossy techniques. However, I may still reserve those results for another work for the purpose of explaining deeper the reasoning, pros, cons, etc. of this type of approach vs. the traditional approach.

---

### Official Review · AnonReviewer3 · 2017-11-27
**An interesting thought but not well justified or tested.**

**Rating:** 4
**Confidence:** 4

**Review:**

The paper proposes to use discrete wavelet transforms combined with downsampling to achieve arguably better pooling output compared to average pooling or max pooling. The idea is tested on small-scale datasets such as MNIST and CIFAR.

Overall, a major issue of the paper is the linear nature of DWT. Unless I misunderstood the paper, linear DWT is being adopted in the paper, and combined with the downsampling and iDWT stage, the transform is linear with respect to the input: DWT and iDWT are by definition linear, and the downsampling can be viewed as multiplying by 0. As a result, if my understanding is correct, this explains why the wavelet pooling is almost the same as average pooling in the experiments (other than MNIST). See figures 10, 12 and 14.

The rest of the paper reads reasonable, but I am not sure if they offset the issue above.

Other minor comments:

- I am not sure if the issue in Figure 2 applies in general to image classification issues. It surely is a mathematical adversarial to the max pooling principle, but note that this only applies to the first input layer, as such behavior in later layers could be offset by switching the sign bits of the previous layer's filter.

- The experiments are largely conducted with very small scale datasets. As a result I am not sure if they are representative enough to show the performance difference between different pooling methods.

---

> ### Author Response · Authors · 2018-01-04
> **Re: An interesting thought but not well justified or tested.**
>
> Linear DWT concerns:
> - Haar wavelet is linear in nature
> - Wavelet are linear, but there also are nonlinear wavelets
> - The linearity of the wavelet doesn't impact its ability to constructively be applied to linear or nonlinear data
> - Average pooling and our method have some overlap in their approach, but differ greatly in execution.
>
> Larger Datasets:
> - I agree that testing on a larger dataset would remove all doubt.
> - I will apply this method to a larger dataset, but I am not sure it will make it into this paper, or be reserved for another manuscript.
>
> We used the Haar wavelet basis as a starting point, a prototype to prove that wavelets could be a viable alternative to the traditional methods. Although Haar is linear as a basis, there are others that are not, and are more advanced in nature. We believe that such a comparison to these bases would be best suitable for another paper or journal article where more depth and discussion could be given on the wavelets themselves, versus the viability of the method.

---

### Public Comment · (anonymous) · 2018-01-25
**Will the code be available anywhere?**

I was just wondering if your code will be available to other researchers at some point? The idea is very interesting

---

> ### Author Response · Authors · 2018-01-25
> **Re: Will the code be available anywhere?**
>
> Short answer is yes. We would love to give access to the code. The longer answer is that it needs to be made more efficient so that the implementation time is reduced. When it was written it wasn't written for CUDA, or MEX, and thus doesn't have the speedups afforded by precompiling, etc. When that happens we will make it available.

---

### Author Response · Authors · 2018-02-19
**FWT vs DWT**

I changed the formula referenced in the paper (for the camera-ready version) to the Fast Wavelet Transform instead of the Discrete Wavelet Transform, to better reflect how our method is implemented. The FWT is faster due to its usage of subband coding and filter banks to recursively create resolutions at higher levels, and reconstruct in a similar manner. It has a complexity of O(N), while the DWT is O(4MN^2log2N), where M is the number of vanishing moments of the wavelet, and N is the number of data points. Sometimes these terms (FWT & DWT) are used interchangeably, and I didn't reference the correct equation.

---

### Public Comment · ~milad_vaali_esfahaani1 · 2018-07-11
**Implementation of backpropagation**

I want to Implement this article.what can i do for implementing back-propagation in python with karas . I want to use keras without changing that. I don't want to write code of back-propagation from scratch.
Is this possible?

---

### Decision · Program_Chairs · 2018-01-29
**ICLR 2018 Conference Acceptance Decision**

**Decision:**

Accept (Poster)

**Comment:**

The idea of using wavelet pooling is novel and will bring many interesting research work in this direction. But more thorough experimental justification such as those recommended by the reviewers would make the paper better. Overall, the committee feels this paper will bring value to the conference.